# ⊕ NIMBA : Towards Robust and Principled Processing of Point Clouds With SSMs

## Abstract

Transformers have become dominant in large-scale deep learning tasks across various domains, including text, 2D and 3D vision. However, the quadratic complexity of their attention mechanism limits their efficiency as the sequence length increases, particularly in high-resolution 3D data such as point clouds. Recently, state space models (SSMs) like Mamba have emerged as promising alternatives, offering linear complexity, scalability, and high performance in long-sequence tasks. The key challenge in the application of SSMs in this domain lies in reconciling the non-sequential structure of point clouds with the inherently directional (or bi-directional) order-dependent processing of recurrent models like Mamba. To achieve this, previous research proposed reorganizing point clouds along multiple directions or predetermined paths in 3D space, concatenating the results to produce a single 1D sequence capturing different views. In our work we introduce a method to convert point clouds into 1D sequences that maintains 3D spatial structure with no need for data replication, allowing Mamba's sequential processing to be applied effectively in an almost permutation-invariant manner. In contrast to other works, we found that our method does not require positional embeddings, and allows for shorter sequence lengths while still achieving state-of-the-art results in ModelNet40 and ScanObjectNN datasets and surpassing Transformer-based models in both accuracy and efficiency.

## 1 Introduction

Today, the transformer architecture is the most common technology powering large-scale deep learning systems. Since their introduction by Vaswani et al. (2017), transformers have been widely adopted in text (Dubey et al., 2024; Team et al., 2023), image (Dosovitskiy et al., 2020; Touvron et al., 2021; Liu et al., 2021), and video (Bertasius et al., 2021; Tong et al., 2022; Liu et al., 2022; Wang et al., 2023) data, as well as in the multimodal setting (Radford et al., 2021; Liu et al., 2024a). In 3D vision and particularly point cloud analysis, transformers achieve state-of-the-art results (Guo et al., 2021; Yu et al., 2022; Pang et al., 2022; Wu et al., 2024), often surpassing convolution-based approaches (Wu et al., 2019; Li et al., 2018) at scale.

While the structure of transformers is favorable on modern hardware, their softmax attention mechanism drastically affects the model complexity with respect to the sequence length (in text) or the number of patches (in image/video/point clouds) – which scales quadratically with respect to these quantities. Over the years, this issue inspired extensive research on alternative *sequence mixer strategies*, such as separable attention (Wang et al., 2020; Choromanski et al., 2020; Lee-Thorp et al., 2021; Chen et al., 2021; Wortsman et al., 2023; Arora et al., 2024; Ramapuram et al., 2024), as well as the development more efficient GPU implementations of softmax attention (Dao et al., 2022; Dao, 2023; Shah et al., 2024). However, the most relevant leap forward on this issue arguably came in recently and coincided with the design of state-space models such as Mamba (Gu & Dao, 2023) as well as other parallelizable token mixers (Poli et al., 2023; De et al., 2024; Yang et al., 2023; Qin et al., 2024; Yang et al., 2024; Beck et al., 2024). SSMs are highly parallelizable RNN-like[1] *sequential* blocks that sparked from the seminal work of Gu et al. (2020; 2022), where complexity

---

[1] Alternatively, SSMs can be seen as fast and well-parametrized linear attention mechanisms (Dao & Gu, 2024a; Sieber et al., 2024; Ali et al., 2024). This connection between RNNs and linear attention dates back to earlier works (Katharopoulos et al., 2020; Schlag et al., 2021).

scales linearly with sequence length, unlocking long-context processing in several challenging applications such as audio (Goel et al., 2022) and DNA modeling (Nguyen et al., 2024). On top of improved efficiency in the long-context setting, Mamba, xLSTMs, and other new RNN/linear attention variants often show general improvements in downstream performance (see empirical study on text in Waleffe et al. (2024)) and reasoning capabilities e.g. in the long-range arena (Tay et al., 2020) and other challenging text tasks where transformers can struggle (Beck et al., 2024).

Along with 2D vision (Liu et al., 2024c; Zhu et al., 2024; Li et al., 2024), Mamba was soon applied to several 3D data domains (Xing et al., 2024; Zhang et al., 2024b) and specifically to point clouds (Liang et al., 2024; Zhang et al., 2024a; Liu et al., 2024b), where it looks particularly promising since datasets such as ScanNet (Dai et al., 2017), often contain over $100k$ points. Compared to the 1D setting such as audio and text, where Mamba (just like any RNN variant) naturally processes data left-to-right or bidirectionally without the need for positional embeddings (Waleffe et al., 2024), 2D and 3D data is inherently not sequential and hence pose an intriguing conceptual challenge to the application of Mamba. Note that instead non-causal attention-based models are *set operations*[2], where position is often added directly to the features (Vaswani et al., 2017; Su et al., 2024) or to the attention matrix (Press et al., 2021). As such, while 1D/2D and 3D data are conceptually similar in attention (i.e., set operation + positional information as a feature, see e.g. Dosovitskiy et al. (2020)), they become puzzling when an *order-sensitive* sequential block processes inputs. This motivates our question, which we explore in the 3D setting:

> *How shall we apply a sequential model to non-sequential data, e.g. a point cloud?*

Addressing this question is scientifically intriguing, timely, and crucial for fully harnessing the potential of new efficient attention variants in the 3D domain. While inspecting the constantly growing literature on Mamba applications in 3D vision, we can see two recurring patterns[3].

(A) 3D point cloud data has to be converted into an *ordered* sequence before Mamba can be applied. This has been achieved with different strategies such as reordering the points along axis and replicating the sequence (Liang et al., 2024; Zhang et al., 2024a) or scanning it from different directions (Liu et al., 2024b).

(B) Much like in transformers, positional embeddings are used. This information is conceptually redundant since (1) positional information is contained in the feature themselves, and (2) the ordering of patches along the constructed sequence is already used by Mamba implicitly. Note that in text, Mamba is often applied without positional embeddings (Waleffe et al., 2024).

While the performance of Mamba in 3D data already shows promise, often surpassing transformers in accuracy and processing speed, points A and B above showcase that applying Mamba to point clouds poses nontrivial challenges, potentially affecting robustness and generalization out of distribution. Towards understanding and improving our understanding of the optimal preprocessing strategies for Mamba-powered models for 3D data, we introduce the following contributions:

1. We draw attention to the problem of sequence construction when applying Mamba to 2D or 3D data. We complement our discussion with both theoretical considerations on invariances and positional embeddings (Sec. 3.3) and ablations (Sec. 4).

2. We introduce NIMBA [4], a Mamba-like model that feeds 3D data points based on an intuitive 3D-to-1D reordering strategy that preserves the spatial distance between points (Sec. 3.3.2). This strategy allows for *safe removal of positional embeddings* without significantly affecting (most times, improving) performance. This is in stark contrast to all previously introduced Mamba strategies in point clouds where our ablation reveal a performance drop when positional embeddings are not used. Along with improved efficiency, our results (Sec. 4) showcase how principled ordering along a point cloud can improve performance of Mamba models in this setting.

---

[2] As noted by Kazemnejad et al. (2024), BERT encoders (Devlin et al., 2019) on text without positional embeddings are a bag-of-words model. Deep causal self-attention (decoders) can instead recover positional information at the second layer.

[3] Similar discussion would hold for 2D data, see e.g. Liu et al. (2024c).

[4] The name NIMBA is derived from the combination of Nimbus (latin for "dark cloud") and Mamba.

3. We show how our ordering strategy in `NIMBA` drastically improves robustness of the model against data transformations such as rotations and jittering (Sec. 4).

We compare our contributions with previous work in Table 1.

| Model | Backbone | Sequence length | Bidirectional | Pos embedding |
|-------|----------|-----------------|---------------|---------------|
| PCT | Transformer | $N$ | × | ✓ |
| PointMAE | Transformer | $N$ | × | ✓ |
| PointMamba | Mamba | $3N$ | × | ✓ |
| Point Cloud Mamba | Mamba | $3N$ | ✓ | ✓ |
| OctreeMamba | Mamba | $N$ | ✓ | ✓ |
| Point Tramba | Hybrid | $N$ | ✓ | ✓ |
| PointABM | Hybrid | $N$ | ✓ | ✓ |
| `NIMBA` (Ours) | Mamba | $N$ | × | × |

Table 1: Comparison of Models based on Architecture, Sequence Length, Directionality and Positional Embedding. We denote with $N$ number of points in the point cloud.

## 2 RELATED WORK

**Point Cloud Transformers.** Transformers, initially designed for NLP, have proven to be highly effective in point cloud analysis due to their global attention mechanisms and permutation invariance properties. Early models like Vision Transformer (ViT) (Dosovitskiy et al., 2020) demonstrated that transformers could outperform CNNs in classification by applying attention directly to patches. This success inspired their application to point clouds, where global feature modeling is crucial.
Point-BERT (Yu et al., 2022) applied BERT's masked modeling to 3D data, using a discrete tokenizer to convert point patches into tokens and self-supervised pre-training to recover masked points. Point-MAE (Pang et al., 2022) further advanced this with masked autoencoding, learning latent representations by reconstructing missing parts from masked inputs. These methods outperformed traditional models by leveraging large unlabeled datasets, but their self-attention's quadratic complexity limits scalability. OctFormer (Wang, 2023) addressed this by using octree-based attention, reducing computational costs through local window partitioning while maintaining performance for large-scale tasks. PointGPT (Chen et al., 2023) uses an autoregressive framework inspired by GPT, treating point patches as sequential data to predict the next patch. This pre-training strategy shows strong generalization in few-shot and downstream tasks, enhancing transformers' flexibility in point cloud processing. Similarly, PCT (Guo et al., 2021) leverages transformer architecture with permutation invariance to process unordered point sequences. By using farthest point sampling and nearest neighbor search, PCT captures local context effectively, achieving state-of-the-art performance in tasks like shape classification and part segmentation.

**Point Cloud State Space Models.** The use of SSMs in point cloud analysis has recently gained attention as a promising approach to address the computational limitations of transformer-based architectures. Although transformers effectively capture global dependencies, their quadratic complexity impedes scaling to high-resolution point clouds. In contrast, SSMs like the Mamba architecture offer linear complexity and efficient long-range modeling. Yet a primary challenge in applying SSMs to point clouds is the unordered nature of the data, which does not align well with the sequential processing of SSMs. To address this, researchers have already proposed methods to convert point clouds into sequences.

One common strategy among recent works is to design ordering methods that preserve the spatial relationships within point clouds when converting them into sequences. For example, Point-Mamba (Liang et al., 2024) and Point Cloud Mamba (PCM) (Zhang et al., 2024a) introduce axis-wise reordering techniques and sequence replication to improve SSMs' ability to capture both local and global structures. Other studies use hierarchical data structures to reflect the spatial hierarchy, such as octree-based ordering in OctreeMamba (Liu et al., 2024b), which organizes points in a $z$-order sequence, maintaining spatial relationships while capturing features at multiple scales. While preserving spatial relationships during serialization is crucial, enhancing local feature extraction within SSMs is equally important for point cloud analysis. Although SSMs model long-range

dependencies efficiently, capturing fine-grained local details remains important. Mamba3D (Han et al., 2024) addresses this by introducing a Local Norm Pooling block to improve local geometric representation. It also employs a bidirectional SSM operating on both tokens and feature channels, balancing local and global structure modeling without increasing computational complexity.

Following (Waleffe et al., 2024; Dao & Gu, 2024a), Another line of research combines the strengths of Transformers and SSMs for better performance and efficiency. PoinTramba (Wang et al., 2024) integrates Transformers to capture detailed dependencies within point groups, while Mamba models relationships between groups using a bidirectional importance-aware ordering strategy. By re-ordering group embeddings based on importance scores, this approach improves performance and addresses random ordering issues in SSMs. SSMs are also applied to point cloud completion and filtering. 3DMambaIPF (Zhou et al., 2024) uses Mamba's selection mechanism with HyperPoint modules to reconstruct point clouds from incomplete inputs, preserving local details often lost in Transformers. For filtering, it combines SSMs with differentiable rendering to reduce noise in large-scale point clouds, improving alignment with real-world structures and handling datasets with hundreds of thousands of points where other methods struggle.

These works show that SSMs effectively address key challenges in point cloud analysis, offering efficient and scalable solutions for various tasks. With advances in serialization, feature extraction, and hybrid architectures, SSMs have become a valuable approach for advancing 3D vision applications. In this work, our goal is to further strengthen these results by offering a simple, principled and robust solution for constructing input sequences input of Mamba-like models.

## 3 MODEL DESIGN

We start in Sec. 3.1 by overviewing the processing strategies common in the point cloud literature. In Sec. 3.2 by recalling the basic properties of Mamba and self attention, highlighting their connections. We continue in Sec. 3.3 by describing how Mamba-like processing of point cloud data leads to interesting considerations around the effects of assigning an order to patches in 3D space. We then analyze the PointMamba strategy in Sec. 3.3.1 and in Sec 3.3.2 we describe our methodology.

### 3.1 BASIC STRATEGIES IN POINT CLOUD ANALYSIS

We outline the typical pipeline used for point cloud analysis in recent deep models. These are not specific to our model, but will allow us to make connections and simplify the discussion.

**Preprocessing.** The goal of the preprocessing phase is to reduce the cardinality of the point cloud while preserving the structure of the data, allowing for more efficient computation in subsequent stages. Formally, let $\mathcal{P} = \{\mathbf{p}_i \mid \mathbf{p}_i \in \mathbb{R}^3, i = 1, \ldots, N\}$ represent the point cloud, where $N$ is the total number of points, and $\mathbf{p}_i = (x_i, y_i, z_i)$ denotes the 3D coordinates of each point. After normalizing the points, a 2-step process is often followed:

1. *Center Selection:* $n_c$ points are selected using the Farthest Point Sampling (FPS) algorithm. FPS iteratively selects points that are farthest from each other, ensuring that the sample is representative of the original point cloud. These points are referred to as "centers" $\{C_i\}_{i=1}^{n_c}$, providing *global* information about the object.

2. *Patch Creation:* For each center, $n_p$ nearest points are selected using the k-Nearest Neighbors (kNN) algorithm. This results in a set of patches $\{P_i\}_{i=1}^{n_c}$, each centered around one of the chosen centers, capturing more localized information about the object.

The values of $n_c$ and $n_p$ are hyperparameters of this preprocessing stage. In our experiments, we followed the procedure of previous work in the literature that can be found in Appendix A.

**Patch Embedding.** Each patch $P_i$ is embedded into a fixed-dimensional vector $\mathbf{p}_i$ through a sequence of expansions, convolutions, and linear projections. This embedding process is a pointwise transformation from $\mathbb{R}^{\text{BS} \times n_p \times 3} \rightarrow \mathbb{R}^{\text{BS} \times n_p \times d_e}$. Here BS is the batch size, $n_p$ is the number of points per patch, and $d_e$ is the embedding dimension. This embedding captures local geometric information within each patch, which is crucial for understanding the finer details of the object structure.

**Center Embedding.** Each center $C_i$ is embedded into a fixed-dimensional vector $\mathbf{c}_i$ to capture global *positional information* and provide context for the relationships between different patches. This embedding process is a pointwise transformation from $\mathbb{R}^{\text{BS} \times n_c \times 3} \rightarrow \mathbb{R}^{\text{BS} \times n_c \times d_e}$.

Following the setting of transformer-based models (Vaswani et al., 2017), the center embedding serves a similar purpose to positional embeddings in point cloud analysis. By focusing on capturing spatial relationships across the entire point cloud, it is believed to provide a complementary view to the local information captured by patch embeddings.

## 3.2 ATTENTION AND MAMBA

In this subsection we denote by $X \in \mathbb{R}^{N \times d}$ a generic input consisting of $N$ elements in $d$ dimensions. In the context of Sec. 3.1, $X$ is the sequence of patch embeddings possibly augmented with positional embeddings. We denote by $X_i$ the $i$-th row of $X$, corresponding to an input token in text or a patch/point cluster in vision. We describe attention and Mamba-like processing of $X$ yielding updated representations $Y \in \mathbb{R}^{N \times d}$.

**Attention.** The standard self-attention block (Vaswani et al., 2017) consists of three matrices: $W_Q$, $W_K$, and $W_V$, which are the learnt parameters of the model. These matrices, when multiplied with the input $X \in \mathbb{R}^{N \times d}$, yield the queries $Q \in \mathbb{R}^{N \times d}$, keys $K \in \mathbb{R}^{N \times d}$, and values $V \in \mathbb{R}^{N \times d}$: $Q = XW_Q, K = XW_K, V = XW_V$. These are combined to produce the output $Y \in \mathbb{R}^{N \times d}$.

$$Y = \text{softmax}\left(\frac{QK^\top}{\sqrt{d}}\right) V, \tag{1}$$

where softmax is applied row-wise. Assuming for simplicity $W_V$ is the identity matrix, we get

$$Y = \Phi_{\text{SDPA}}^X \cdot X, \tag{2}$$

where $\Phi_{\text{SDPA}} \in \mathbb{R}^{N \times N}$ mixes tokens as follows:

$$\Phi_{\text{SDPA}}^X = \text{softmax} \begin{pmatrix} \frac{1}{\sqrt{d}}X_0 W_Q W_K^\top X_0^\top & \frac{1}{\sqrt{d}}X_0 W_Q W_K^\top X_1^\top & \cdots & \frac{1}{\sqrt{d}}X_0 W_Q W_K^\top X_N^\top \\ \frac{1}{\sqrt{d}}X_1 W_Q W_K^\top X_0^\top & \frac{1}{\sqrt{d}}X_1 W_Q W_K^\top X_1^\top & \cdots & \frac{1}{\sqrt{d}}X_1 W_Q W_K^\top X_N^\top \\ \vdots & \vdots & \ddots & \vdots \\ \frac{1}{\sqrt{d}}X_N W_Q W_K^\top X_0^\top & \frac{1}{\sqrt{d}}X_N W_Q W_K^\top X_1^\top & \cdots & \frac{1}{\sqrt{d}}X_N W_Q W_K^\top X_N^\top \end{pmatrix}. \tag{3}$$

In causal self-attention, used e.g. in language modeling, the upper triangular portion of $\Phi_{\text{SDPA}}$ is set to 0. For vision application, $\Phi_{\text{SDPA}}$ is often used without masking.

**Mamba.** Architectures based on state-space models (SSMs) (Gu et al., 2022; Gu & Dao, 2023; Dao & Gu, 2024a) compute the output $Y$ through a dynamic recurrence of input signals. $X$ is seen as a time-series where time flows from left to right: $X_1, X_2, \ldots, X_N$. Starting from $Z_{i-1} = 0 \in \mathbb{R}^n$

$$Z_i = A_i Z_{i-1} + B_i X_i \tag{4a}$$
$$Y_i = C_i Z_i + D_i X_i, \tag{4b}$$

where $Z_i$ is the hidden state of the system, and the dynamic matrices of appropriate dimensions $A_i, B_i, C_i, D_i$ are functions of the model parameters as well as the input. The S6 block (Gu & Dao, 2023; Dao & Gu, 2024a) parametrizes the recurrence as

$$A_i = e^{-\Delta_i W_A}, \qquad B_i = \Delta_i W_B X_i, \qquad C_i = W_C X_i, \qquad D_i = W_D X_i \tag{5}$$

and $\Delta_i = \text{softplus}(W_\Delta X_i + b_\Delta)$, with $W_\Delta, W_A, W_B, W_C, W_D$ are learnt *matrices* of appropriate dimensions, and $b_\Delta$ is a learnt bias. It is well known (Katharopoulos et al., 2020; Ali et al., 2024; Dao & Gu, 2024a; Sieber et al., 2024) that this system can be cast into an attention matrix representation [5], also known in the SSM literature as *convolutional* representation (Gu et al., 2021):

$$Y = \Phi_{\text{S6}}^X \cdot X, \tag{6}$$

where

$$\Phi_{\text{S6}}^X = \begin{pmatrix} C_0 B_0 + D_0 & 0 & \cdots & 0 \\ C_1 A_1 B_0 & C_1 B_1 + D_1 & \cdots & 0 \\ \vdots & \ddots & \ddots & \vdots \\ C_N \prod_{k=1}^{N} A_k B_0 & \cdots & C_N A_N B_{N-1} & C_N B_N + D_N \end{pmatrix}. \tag{7}$$

[5] In modern variants of Mamba such as Mamba2 (Dao & Gu, 2024a), the hidden dimension of $Z$ in Eq. 4 is such that $\Phi_{\text{S6}} \in \mathbb{R}^{N \times N}$. For earlier variants, the transformation is conceptually similar but has to be written in a slightly different form.

**Architecture.** Attention and S6 layers are often used in deep networks by interleaving with MLPs, normalization components and skip connections. We use in this paper the backbone of the Mamba architecture (Gu & Dao, 2023), and refer to the original paper for details as well as to our appendix.

### 3.3 POSITIONAL EMBEDDINGS AND ARROW OF *time* IN MAMBA AND ATTENTION

There are two crucial macroscopic differences between $\Phi_{SDPA}$ and $\Phi_{S6}$:

- $\Phi_{S6}$ is lower triangular, while $\Phi_{SDPA}$ is not.
- $\Phi_{SDPA}$ has an *isotropic* structure: entries close to the diagonal are computed similarly to entries far from the diagonal. Instead, in $\Phi_{S6}$ the distance to the diagonal affects computation: it affects in the number of $A_i$s multiplied together in the formula for each entry.

This divergence between Mamba and Softmax attention is quite deep, and implications are strictly related to the two propositions below:

**Proposition 1** (Softmax Attention). $Y = \Phi_{SDPA}^{X} \cdot X$ *is invariant to row-wise permutations* $\Pi$ *of the input. For all* $X, \Pi$ *and model parameters, we have* $\Phi_{SDPA}^{\Pi(X)} \cdot \Pi(X) = \Pi(\Phi_{SDPA}^{X} \cdot X)$.

**Proposition 2** (Mamba). $Y = \Phi_{S6}^{X} \cdot X$ *is **not invariant** to row-wise permutations* $\Pi$ *of the input: there exists* $X, \Pi$ *and model parameters such that* $\Phi_{S6}^{\Pi(X)} \cdot \Pi(X) \neq \Pi(\Phi_{S6} \cdot X)$.

Proposition 1 directly follows from the fact that attention is a *set operation* (Vaswani et al., 2017), and proposition 2 is also easy to prove (see appendix).

**Pros of being sequential.** Cirone et al. (2024) proved that S6 – with no need for positional embeddings – can simulate any autonomous nonlinear dynamical system evolving in the direction $i \to i+1$. This result is rooted in more general statements regarding Turing Completeness of RNNs (Siegelmann & Sontag, 1992; Chung & Siegelmann, 2021). Indeed, in language modeling, Mamba is used *without positional embeddings* (Waleffe et al., 2024), in contrast to Softmax Attention without masking which requires positional embedding information capture distance information within $X$[6].

**Cons of being sequential.** While in the text is convenient to drop positional embeddings, in the 2D and 3D applications, the notion of "position" cannot be easily captured by 1D ordering in a sequence: when processing data $X$ where each $X_i$ relates to a precise position in space, the output $Y$ crucially depends on the chosen order the $X_i$s are arranged into – in contrast with softmax attention (see propositions above). In a point cloud, we might order along the principal axis in 3D space and feed point clusters one at a time along these axes (Liang et al., 2024) or along an octree-determined path (Liu et al., 2024b). The output of S6, in this case, still depends on the processing order, regardless of the inclusion of additional positional embeddings in $X$ and despite the potential bidirectional application of such models.

In this paper, our goal is to work towards a principled strategy for processing point clouds with inherently sequential models such as Mamba. We first describe in-depth one existing approach (Liang et al., 2024) in Sec. 3.3.1 and then propose a patch reordering strategy that is able to match or improve performance compared to existing approaches, without requiring positional embedding but *relying solely on the sequential patch ordering* we introduce. This both reveals sensitivity to Mamba in sequence construction and potential for future developments using our strategy.

### 3.3.1 POINTMAMBA STRATEGY

Despite the conceptual difficulty in processing 3D data with sequential models, several approaches have been tested in the the literature (see Sec. 2). We here present the strategy proposed by Point-Mamba (Liang et al., 2024): Following the notation of Sec. 3.1, centers $\{C_i\}_{i=1}^{n_c}$ are first sorted along each axis $(x, y, z)$ independently, resulting in three separate *ordered* sequences: $(C_i^x)_{i=1}^{n_c}$, $(C_i^y)_{i=1}^{n_c}$, and $(C_i^z)_{i=1}^{n_c}$. For each axis-sorted sequence, we obtain the corresponding patch embeddings $\{\mathbf{p}_i\}$ and center embeddings $\{\mathbf{c}_i\}$, which are then concatenated in the three orders above to form the input sequence $X$.

---

[6]Kazemnejad et al. (2024) recently proved that causal self-attention can instead recover positional information in 1D structures. Yet, modern practice still adopts positional embeddings by default also in this setting.

Figure 1: Ordering Strategy of `NIMBA` and PointMamba.

This strategy allows for successful processing, as we report in Sec. 4, yet has several weaknesses:

- The method results in the sequence length being tripled, introducing redundancy and negatively affecting efficiency.

- Centers that are close in the 3D space may not be adjacent in the sequence, which can affect the model's robustness and ability to capture spatial relationships effectively.

- As we show in Sec. 4, this method is highly sensitive to the presence of positional embedding information. While this is common in standard attention-based architectures, it is less natural in Mamba-based models. In addition, it increases number of parameters and introduces additional redundancy.

### 3.3.2 NIMBA STRATEGY

To overcome the limitations of the PointMamba strategy, we propose the **NIMBA** approach, designed to better maintain geometric relationships by ensuring that consecutive centers in the 1D sequence fed to the model are close in 3D space. The `NIMBA` strategy is based on the concept of local proximity preservation:

1. *Initial Axis-Wise Ordering:* Initially, centers are flattened by ordering along the y-axis. Any initial order could work, but we chose the y-axis as empirical results indicated that this ordering reduces the computational cost of the following phase.

2. *Proximity Check:* We scan the sequence obtained previously and we iteratively check the distance between the current and the next center. If the distance exceeds a predefined threshold $r$, we look for a center along the sequence that is near enough to the starting center and place it next to it. If no suitable center is found within the threshold, the sequence proceeds to the next center without modification. In this way, we ensure that consecutive centers in the sequence have a distance less than $r$.

Mathematically, the proximity check can be described as $\|C_i - C_{i+1}\| < r$, where $\|\cdot\|$ represents the Euclidean distance in 3D space. The choice of $r$ is crucial: a high threshold value (e.g., $r \geq 2\sqrt{3}$, the diagonal of a unit cube) means that the sequence remains similar to the initial order since the requirement will always be satisfied. Instead, using a low threshold (e.g., $r = 0$) is computationally expensive since each center would be compared to all the others in the sequence, and will result in an ordering identical to the initial axis-wise order, as no centers will be considered close enough to trigger reordering. In our experiment we found $r = 0.8$ to be a good balance between quality of reordered sequence and computational cost. Other than efficiency reasons, the choice of the threshold r is related to the nature of point cloud datasets. Indeed, in ModelNet and Scanobject datasets the objects are contained in a $[-1, 1]^3$ cube. The literature confirms that this comes from the nomalization step, which is a standard procedure in similar works and datasets. The choice of r can be interpreted as a portion of the distance between the center of the scene and the border of the scene, which should be $40\%$. Figure 2 show the complete pipeline of `NIMBA`

Figure 1 show the sequence created from the two strategies. When comparing to the PointMamba approach, `NIMBA` does not rely on positional embeddings and doesn't replicate the sequence: the reordering strategy proposed leverages the spatial relationships of points, allowing the model to rely only on the patch embedding, thus enhancing accuracy and stability.

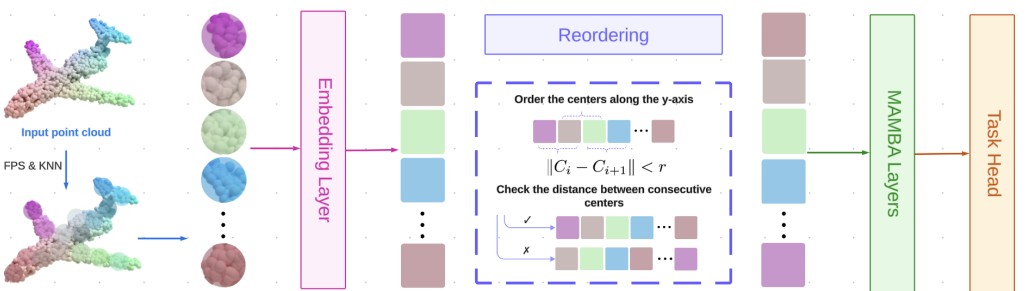

Figure 2: Overview of `NIMBA` pipeline.

# 4 EXPERIMENTAL RESULTS

To make comparisons as fair as possible, we implemented our model in the code environment of PointMamba (Liang et al., 2024), which in turn builds on the Point-MAE framework (Pang et al., 2022)[7]. Rather than fine-tuning a pre-trained model, we trained from scratch to better highlight the difference between each setting. In Appendix A, we provide a comprehensive dataset description as well as implementation details for `NIMBA`. We report the grid search process used to select the learning rates for reproducing experiments from other models. We dedicate to all methods we reproduce the same tuning efforts as `NIMBA`, and repeat all experiments three times and report the results as mean accuracy ± standard deviation.

## 4.1 OBJECT CLASSIFICATION

We evaluate our proposed model, `NIMBA`, against various baseline models on multiple object classification benchmarks, including ModelNet (Wu et al., 2015) and three versions of ScanObjectNN (OBJ-BG, OBJ-ONLY, and PB-T50-RS from Uy et al. (2019)). Results are summarized in Table 2.

| Model | Backbone | Param. (M)↓ | Accuracy (%)↑ | | | |
|---|---|---|---|---|---|---|
| | | | **ModelNet** | **OBJ-BG** | **OBJ-ONLY** | **PB-T50-RS** |
| PointNet (Qi et al., 2017a)* | Neural Network | 3.5 | 89.2 | 73.3 | 79.2 | 68.8 |
| PointNet++ (Qi et al., 2017b)* | Neural Network | 1.5 | 90.7 | 82.3 | 84.3 | 77.9 |
| PCT (Guo et al., 2021)* | Transformer | 2.9 | 90.17 | - | - | - |
| Point Mamba[†] | Mamba | 12.3 | $92.08 \pm 0.16$ | $87.80 \pm 0.72$ | $87.20 \pm 0.88$ | $82.20 \pm 0.45$ |
| **NIMBA (Ours)** | Mamba | 12.3 | $\mathbf{92.10 \pm 0.14}$ | $\mathbf{89.06 \pm 0.42}$ | $\mathbf{89.29 \pm 0.23}$ | $\mathbf{83.91 \pm 0.38}$ |
| Point-MAE[†] | Transformer | 22.1 | $92.30 \pm 1.02$ | $86.77 \pm 0.91$ | $86.83 \pm 0.78$ | $81.23 \pm 0.77$ |
| PointMamba[†] | Mamba | 23.86 | $92.08 \pm 0.19$ | $88.01 \pm 0.77$ | $86.49 \pm 0.49$ | $83.01 \pm 0.82$ |
| **NIMBA (Ours)** | Mamba | 23.86 | $\mathbf{92.10 \pm 0.14}$ | $\mathbf{89.80 \pm 0.36}$ | $\mathbf{89.76 \pm 0.37}$ | $\mathbf{84.21 \pm 0.65}$ |

Table 2: Accuracy on classification tasks. Different scales are reported. * are values reported from the PointMamba paper (Liang et al., 2024), while [†] are our reproducing choosing the best-performing learning rate for each model and task.

**Transformer-based Models.** Transformer-based models such as PCT and Point-MAE achieve competitive accuracies on these benchmarks. However, `NIMBA` surpasses these models by up to ≈ 2% on several datasets while using fewer parameters. Importantly, `NIMBA` achieves these improvements without employing positional embeddings.

**Mamba-based Models.** For Mamba-based architectures, our baseline PointMamba achieves strong performance. `NIMBA` exceeds PointMamba across all datasets, with accuracy improvements of up to 1.5%. Additionally, when scaling up to 23.86M parameters, `NIMBA` continues to enhance its performance, surpassing the larger PointMamba model. These results demonstrate that `NIMBA` effectively leverages additional parameters to improve accuracy while maintaining efficiency.

**Training Efficiency.** Beyond accuracy, we also assess the training efficiency of `NIMBA` compared to PointMamba. As shown in Table 3, `NIMBA` reduces the training time by ≈ 14% on ModelNet and ≈ 17% on ScanObjectNN after 300 epochs of training, highlighting the efficiency of our model.

---

[7]Code will be released after publication, along with a github repository.

Overall, NIMBA outperforms both transformer-based and Mamba-based baseline models without relying on positional embeddings, demonstrating its effectiveness in object classification tasks.

| Models | Param.(M)↓ | Time (m)↓ | |
| --- | --- | --- | --- |
| | | ModelNet | ScanObjectNN |
| PointMamba[†] | 17.4 | 500 | 240 |
| **NIMBA (Ours)** | **17.4** | **430** | **200** |

Table 3: Training time comparison in 300 epochs

## 4.2 PART SEGMENTATION

We evaluate NIMBA on the part segmentation task using the ShapeNetPart dataset. As shown in Table 4, we report the mean IoU (mIoU) for both class-level (Cls.) and instance-level (Inst.) metrics. NIMBA achieves higher Cls. mIoU compared to both Transformer-based and Mamba-based models and demonstrates competitive performance in Inst. mIoU. Specifically, NIMBA outperforms our Mamba-based baseline, Point-Mamba, by $\approx 1\%$ in Cls. mIoU while maintaining similar Inst. performance. Since all models are tuned to best independently, we attribute this performance boost to our improved reordering strategy.

| Models | Param.(M)↓ | Cls. mIoU(%)↑ | Inst. mIoU(%)↑ |
| --- | --- | --- | --- |
| PointNet[*] | - | 80.39 | 83.7 |
| PointNet++[*] | - | 81.85 | 85.1 |
| Point-MAE[†] | 27.1 | $83.91 \pm 0.43$ | $85.7 \pm 0.23$ |
| PointMamba[†] | 17.4 | $83.37 \pm 0.17$ | $85.07 \pm 0.12$ |
| **NIMBA (Ours)** | **17.4** | $\mathbf{84.36 \pm 0.06}$ | $\mathbf{85.54 \pm 0.05}$ |

Table 4: Performance comparison on the ShapeNetPart segmentation task. [*] indicates values reported in the PointMamba paper (Liang et al., 2024), while [†] denotes our reproduced results using the best-performing learning rates for each method.

## 4.3 ABLATIONS

Here, we present a series of ablation studies to investigate the impact of the different components even further. In particular, in Sec. 4.3.1 we show and compare the effects of positional embedding, in Sec. 4.3.2 we test the robustness of models when augmentation and noise are applied and in Sec. 4.3.3 we see how a bidirectional implementation of Mamba affects performances. All the ablations were made on the classification task on the ScanObjectNN dataset OBJ-BG variation.

### 4.3.1 EFFECT OF POSITIONAL EMBEDDING

To investigate the impact of positional embedding (PE), we conducted a series of experiments comparing transformer, Mamba, and hybrid models. As shown in Table 5, performance generally declines when PE is removed, affecting both models with attention blocks and Mamba-based models. This includes PoinTramba, which, despite outperforming NIMBA under normal conditions, relies heavily on PE. Without it, NIMBA achieves better results. We observed that many Mamba-like models using PE often replicate sequences or add bidirectionality to maintain performance. We hypothesize this is due to redundancy: when sequences are insufficiently meaningful, the model scans them multiple times for better information retrieval. In contrast, NIMBA 's reordering strategy preserves sequence length and performs well without PE.

| Models | Acc. with PE(%)↑ | Acc. without PE(%)↑ | Gap(%)↓ |
| --- | --- | --- | --- |
| Point-MAE[†] | $86.77 \pm 0.91$ | $80.24 \pm 0.87$ | $6.53 \pm 1.78$ |
| PointMamba[†] | $87.80 \pm 0.72$ | $83.69 \pm 0.76$ | $4.11 \pm 1.48$ |
| PoinTramba[†] | $92.42 \pm 0.48$ | $86.46 \pm 0.34$ | $5.96 \pm 0.82$ |
| **NIMBA (Ours)** | $\mathbf{89.80 \pm 0.36}$ | $\mathbf{88.12 \pm 0.54}$ | $\mathbf{1.68 \pm 0.90}$ |

Table 5: Influence of positional emebedding (PE) on performance. [*] indicates values reported in the PointMamba paper (Liang et al., 2024), while [†] denotes our reproduced results using the best-performing learning rates.

### 4.3.2 ROBUSTNESS

To further explore the differences between PointMamba and NIMBA , we tested both models by applying the following noise injections to the input point clouds:

- Rotation: A random 3D rotation of the object;

- Random Horizontal Flip (RHF): A random flip along the horizontal axis;

- Jittering: Points in the point cloud are perturbed with Gaussian white noise;

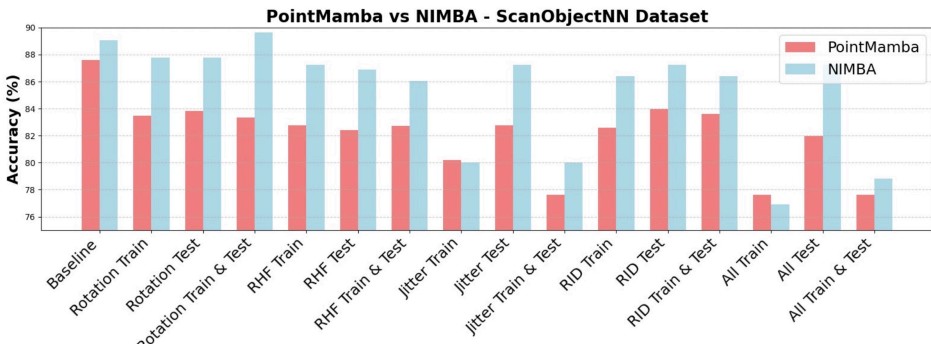

Figure 3: Results of applying noise to the training set, test set, or both. NIMBA demonstrates greater robustness compared to PointMamba, particularly when the noise does not alter the spatial distances between points, such as in the case of rotation.

- Random Input Dropout (RID): Points are randomly removed with a probability p;
- All: A combination of all the noise types listed above.

Each type of noise was applied to the training set, the test set, or both. As shown in Figure 3, NIMBA generally exhibits greater robustness to noise, particularly in the case of rotation, where we even observe an improvement in performance. This confirms that the reordering strategy employed by NIMBA is resilient to noise that preserves pairwise distances between points, such as a rotation.

### 4.3.3 HYDRA

Building on previous works that utilize scanning different directions (Zhang et al., 2024a; Liu et al., 2024b; Wang et al., 2024), we explored the impact of replacing the Mamba block with Hydra (Hwang et al., 2024), a bidirectional extension of Mamba using a quasiseparable matrix mixer, in both PointMamba and

| Models | Param.(M)↓ | Accuracy(%)↑ |
|---|---|---|
| PointMamba with hydra | 12.85 | 86.23 |
| **NIMBA with hydra** | **12.85** | **86.4** |

Table 6: Results when substituting the Mamba block with the Hydra block in the architecture

our NIMBA . Hydra scans sequences in both directions simultaneously, meaning PointMamba still processes a sequence of length $3N$, while NIMBA still processes length $N$. As shown in Table 6, performance generally declined in both cases, likely due to the shift to Hydra, which is based on Mamba2 (Dao & Gu, 2024b). We recommend future research to focus on optimizing Mamba2 in these contexts, as optimization remains a key challenge with such models.

## 5 CONCLUSION AND FUTURE WORK

In this paper, we introduced NIMBA , a robust and principled approach for point cloud processing using state space models (SSMs). When using such causal models, a key challenge is to effectively convert a 3D set of data into a 1D sequence for proper analysis. We addressed this by proposing a spatially-aware reordering strategy that preserves spatial relationships between points. Differently from others, our method eliminates the need for positional embeddings and sequence replication, enhancing both efficiency and performance. Our experimental results demonstrate that NIMBA outperforms or matches transformer-based and other Mamba-based models on benchmark datasets such as ModelNet, ScanObject, and ShapeNetPart in classification and segmentation tasks.

**Limitations and Future Work.** While NIMBA successfully addresses several challenges in point cloud analysis with SSMs, certain limitations remain. From an optimization standpoint, the model shows limited improvement when scaled. Additionally, when replacing the Mamba block with Mamba2 or integrating NIMBA into hybrid architectures, we observed performance declines, suggesting potential integration issues. We encourage further investigation into optimizing SSMs and leveraging their integration with transformer architecture for point cloud analysis. We believe this work offers a new perspective on applying non-transformer models in domains beyond natural language processing, highlighting the potential of SSMs in 3D vision applications.

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

# A   APPENDIX

## A.1   DATASETS

In our experiments, we evaluate the performance of our model using three publicly available 3D datasets: ModelNet40, ScanObjectNN and ShapeNetPart.

**ModelNet40** (Wu et al., 2015): It is a widely used benchmark synthetic dataset used to evaluate 3D object classification models. It consists of 12,311 CAD models across 40 categories, representing clean and noise-free 3D shapes, such as airplanes, chairs, and cars, offering a diverse set of 3D shapes.

**ScanObjectNN** (Uy et al., 2019): It presents a more challenging real-world scenario by offering around 15,000 objects across 15 categories, scanned from real-world indoor scenes. Unlike the controlled environment of CAD datasets, the objects in ScanObjectNN are captured in cluttered and noisy environments, introducing additional complexity due to background noise, occlusion, and deformation. The diversity of real-world settings in this dataset makes it particularly suited for evaluating the robustness of models in practical object classification tasks. The dataset comes in three variants with different degree of difficulty:

- OBJ-ONLY: the easiest variant in which there is only the object in the scene. This is the most similar variant to a CAD analogous.
- OBJ-BG: an intermediate variant in which there is also the background. We focused mostly on this variant since it is the most similar to the real world.
- PB-T50-RS: the hardest version that adds some perturbations to the objects and can be used as a benchmark to test the robustness on the classification task

**ShapeNetPart** (Yi et al., 2016) It is a widely recognized benchmark for 3D shape segmentation tasks. This dataset is a subset of the larger ShapeNet repository and includes 31,693 3D CAD models categorized into 16 common object classes such as chairs, planes, and tables. Each model is richly annotated with detailed geometric and semantic labels, providing valuable information for training and evaluating segmentation algorithms.

## A.2   TRAINING DETAILS

In this section, we give more details on the training setting that we used for our experiments.

To make a fair comparison, we followed the work proposed by PointMamba  (Liang et al., 2024), PointMAE  (Pang et al., 2022) and PCT (Guo et al., 2021) and we show the specific settings for the 3 different datasets and tasks: object classification on the synthetic ModelNet40 in Table 7, object classification on ScanObjectNN in Table 8 and segmentation on ShapeNet in Table 9. We used a model of dimension 384 with 12 encoder layers and 6 heads across all experiments while there are some slight differences in the number of points sampled, number of patches and number of points per patch across the experiments, but still following the works mentioned above.

The main hyperparameter that we tuned in all the experiments reproduced is the learning rate and we did it with the following criteria:

1. We first did a wide grid search in the range $[0.3 - 0.00001]$ with the values

$$[0.3, 0.1, 0.03, 0.01, 0.003, 0.001, 0.0003, 0.0001, 0.00003, 0.00001]$$

   and took the best value as $lr^*$. This search has the nice property of having a scale 3 factor between each value and gives the order of magnitude the learning rate should have.

2. With $lr*$ we created a new grid search with the values

$$[3lr^*, 2lr^*, lr^*, \frac{lr^*}{2}, \frac{lr^*}{3}]$$

   and took the best value as $newlr^*$. This search, other than fine-graining the choice of best learning rate, also ensures that $lr^*$ is not a boundary value.

3. We checked if $newlr^*$ was equal to $lr^*$ and if not we went back to the previous point and used $newlr^*$ as $lr^*$. We kept up doing so until $newlr^*$ was equal to $lr^*$

4. We then ran the experiment 3 times with $newlr^*$ with 3 different seeds in all our experiments, and report the mean accuracy $\pm$ standard deviation.

| Configuration | Details | Value |
|---|---|---|
| Model Configuration | Transformer Dimension | 384 |
| | Num. of Encoder Layers | 12 |
| | Num. of heads | 6 |
| Points Configuration | Num. of Points | 1024 |
| | Num. of Patches | 64 |
| | Num. of Point per Patches | 32 |
| Training settings | Optimizer | AdamW |
| | Learning Rate | 1e-4 |
| | Weight Decay | 5e-2 |
| | Scheduler Type | Cosine |
| | Num. of Epochs | 300 |
| | Num. of Warm-up Epochs | 10 |
| | Batch Size | 32 |
| | Seeds | 0, 123, 777 |

Table 7: Training configuration for classification on ModelNet40

| Configuration | Details | Value |
|---|---|---|
| Model Configuration | Transformer Dimension | 384 |
| | Num. of Encoder Layers | 12 |
| | Num. of heads | 6 |
| Points Configuration | Num. of Points | 2048 |
| | Num. of Patches | 128 |
| | Num. of Point per Patches | 32 |
| Training settings | Optimizer | AdamW |
| | Learning Rate | 5e-4 |
| | Weight Decay | 5e-2 |
| | Scheduler Type | Cosine |
| | Num. of Epochs | 300 |
| | Num. of Warm-up Epochs | 10 |
| | Batch Size | 32 |
| | Seeds | 0, 123, 777 |

Table 8: Training configuration for classification on ScanObjectNN

# B  MISSING PROOFS

## B.1  PROOF OF PROPOSITION 2

The general formula describing S6 computation is $Y_k = C_k \sum_{j=0}^{k} (\prod_{k=j+1}^{k} A_k) B_k X_j$. Let us pick $N = 2$, we have $Y_0 = C_0 B_0 X_0$ and $Y_1 = C_1 A_1 B_0 X_0 + C_1 B_1 X_1$. Let $\Pi$ swap the first and second inputs. For the reversed sequence, we have $\hat{Y}_0 = \hat{C}_0 \hat{B}_0 X_1$ and $\hat{Y}_1 = \hat{C}_1 \hat{A}_1 \hat{B}_0 X_1 + \hat{C}_1 \hat{B}_1 X_0$. We have to prove that for any realized value of $A, B, C, \hat{A}, \hat{B}, \hat{C}$, there exists a sequence $X$ such that $Y_1 \neq \hat{Y}_0$, i.e. $C_1 A_1 B_0 X_0 + C_1 B_1 X_1 \neq \hat{C}_0 \hat{B}_0 X_1$. It is clear that converse would imply $C_1 A_1 B_0 X_0 = (\hat{C}_0 \hat{B}_0 - C_1 B_1) X_1$, i.e. a strong relationship between the values of $X_0$ and $X_1$.

| Configuration | Details | Value |
|---|---|---|
| Model Configuration | Transformer Dimension | 384 |
| | Num. of Encoder Layers | 12 |
| Points Configuration | Num. of Points | 2048 |
| | Num. of Patches | 128 |
| | Num. of Point per Patches | 32 |
| Training settings | Learning Rate | 1e-4 |
| | Weight Decay | 5e-2 |
| | Scheduler Type | Cosine |
| | Num. of Epochs | 300 |
| | Num. of Warm-up Epochs | 10 |
| | Batch Size | 16 |
| | Seeds | 42, 123, 777 |

Table 9: Training configuration for classification on ShapeNetPart

