# OpenReview forum: "NIMBA : Towards Robust and Principled Processing of Point Clouds With SSMs"
_ICLR.cc/2025/Conference — Submitted to ICLR 2025_

### Official Review · Reviewer_NaDZ · 2024-10-26

**Soundness:** 4
**Presentation:** 3
**Contribution:** 3
**Rating:** 3
**Confidence:** 4

**Summary:**

The paper introduces NIMBA, a new method for point cloud analysis using state-space models (SSMs). The key contribution is a new strategy that converts 3D point clouds into 1D sequences while preserving spatial structure, eliminating the need for positional embeddings. NIMBA builds on the Mamba architecture to improve efficiency and accuracy in tasks like object classification and segmentation. The authors demonstrate that NIMBA outperforms both transformer-based and other SSM-based methods on multiple datasets such as ModelNet40 and ScanObjectNN, showing enhanced robustness to noise and spatial transformations.

**Strengths:**

1. The paper proposes a novel method for converting 3D point clouds into sequences without replication or positional embeddings.
2. Significance: NIMBA achieves state-of-the-art results on multiple datasets, outperforming both transformer and SSM-based baselines.
3. The methodology is well-structured, with clear comparisons to prior work, though some sections could be streamlined for better readability.

**Weaknesses:**

1. Scaling limitations: The model shows limited improvement when scaled, suggesting potential optimization challenges. It would be better if they could verify the effectiveness on larger point cloud datasets, like nuScenes, and Waymo.
2. Performance declines were observed when replacing the Mamba block with the Hydra block, indicating possible limitations in hybrid architectures.
3. some typos should pay attention to, e.g.,
  a) line 099: positional emebddings might be positional embeddings;
  b) line 352: flattered by ordering might be flattened by ordering;
  c) line 377: environment might be environment
  d) line 523: conclusion and Fig. 2 show
  and some formatting issues

**Questions:**

1. Could the authors provide more intuition behind the choice of the proximity threshold of 0.8?
2. Have the authors considered alternative strategies for scaling the model to larger datasets, like nuScenes and Waymo?
3. Hybrid models: what are the potential avenues to optimize NIMBA when integrated with hybrid architectures like Hydra?

**Details Of Ethics Concerns:**

No ethical concerns were identified for this work.

---

> ### Author Response · Authors · 2024-11-25
>
> We thank the reviewer for their comments on our submission:
>
> As also stated in our reply to reviewer 9zEE, our results are in line with similar works. The nature of the field and tasks of **3D vision reveal scaling issues** that apply to all other models and are **deeply connected to saturation of performance in lack of pretraining**. However, we want to highlight that we do not discuss the scaling and pretraining properties of Mamba models, but instead, **we focus on a principled preprocessing strategy that enhances its robustness**.
>
> As also stated in our reply to reviewers 8URG and 9zEE, the threshold **r is a portion of the 3D space in which the point clouds are inserted**. Our value of 0.8 is a balance between gain in performance and computation costs.
>
> We thank the reviewer for pointing out the typos in our research paper. We made the corrections accordingly.

---

### Official Review · Reviewer_9zEE · 2024-10-31

**Soundness:** 2
**Presentation:** 4
**Contribution:** 2
**Rating:** 3
**Confidence:** 5

**Summary:**

The paper introduces a novel method for processing 3D point clouds using State Space Models (SSMs), specifically the Mamba model. The key innovation is a strategy to convert 3D point clouds into 1D sequences that preserves the spatial structure without requiring data replication, thus enabling efficient sequential processing by Mamba in a permutation-invariant manner. The authors claim that their method surpasses Transformer-based models in accuracy and efficiency and does not require positional embeddings. The paper reports state-of-the-art results on ModelNet40 and ScanObjectNN datasets and demonstrates improved robustness against data transformations such as rotations and jittering.

**Strengths:**

1、The linear complexity of SSMs like Mamba, as opposed to the quadratic complexity of Transformer models, makes NIMBA highly scalable for high-resolution 3D data.

2、The paper shows that NIMBA is more robust to data transformations such as rotations and jittering, which is crucial for real-world applications where data can be subject to various distortions.

3、The elimination of positional embeddings and sequence replication makes the model more principled and less reliant on artificial constructs for sequence ordering.

**Weaknesses:**

1、One of the contributions proposed in this paper is the reordering strategy. However, this serialization strategy should be compared and discussed with the Point Cloud Mamba[1]. In the Point Cloud Mamba, many methods are discussed and compared, but they are all missing in this paper. There is even no specific discussion and comparison in the ablation studies. Furthermore, the performance of NIMBA's reordering strategy is dependent on the choice of the threshold parameter, which may require careful tuning for different datasets.

2、The paper notes that NIMBA shows limited improvement when scaled, suggesting that there may be optimization challenges that need to be addressed. There is a noted decline in performance when integrating NIMBA with Mamba2 or in hybrid architectures, indicating potential issues with model integration.

3、Comparative Analysis: The paper primarily compares NIMBA with Mamba-based models; a more comprehensive comparison with other state-of-the-art methods, especially those using different SSMs [1], [2], could provide a fuller picture of NIMBA's performance. Furthermore, this paper claims to surpass the transformer-based method, but it lacks many comparisons with such methods, such as PointBert[3], PointM2AE[4].

4、 How was the threshold parameter determined, and how sensitive is the model's performance to changes in this parameter? How does the removal of positional embeddings affect the model's interpretability, and can the learned representations be easily understood?

5、While the paper claims efficiency improvements, are there specific computational cost analyses, especially for large-scale datasets?

[1] Zhang T, Li X, Yuan H, et al. Point could mamba: Point cloud learning via state space model[J]. arXiv preprint arXiv:2403.00762, 2024.

[2] Han X, Tang Y, Wang Z, et al. Mamba3d: Enhancing local features for 3d point cloud analysis via state space model[J]. arXiv preprint arXiv:2404.14966, 2024.

[3] Yu X, Tang L, Rao Y, et al. Point-bert: Pre-training 3d point cloud transformers with masked point modeling[C]//Proceedings of the IEEE/CVF conference on computer vision and pattern recognition. 2022: 19313-19322.

[4] Zhang R, Guo Z, Gao P, et al. Point-m2ae: multi-scale masked autoencoders for hierarchical point cloud pre-training[J]. Advances in neural information processing systems, 2022, 35: 27061-27074.

**Questions:**

Refer to weakness part.

---

> ### Author Response · Authors · 2024-11-25
>
> We thank the reviewer for pointing out these issues with our submission:
>
> Our results, scales and hyperparameter tunings are in line with other papers. **Scaling issues** apply to all other models and we strongly believe that **are rooted in a saturation of performance in lack of pretraining**. Indeed, gains in performance when scaling models is something very specific to language modeling and leveraging pretraining might not hold in the specific setting of point clouds. However, **we do not discuss the successful scaling and pretraining of mamba models, but instead their robustness and preprocessing strategy**. In fact, we want to highlight that the scope of this paper is to have proofs and insights on how to properly transverse an unordered data modality such as point clouds, with a causal model that needs a sequence as an input. Our experiments show how a proper reordering strategy is the main driver for performance, allowing it to be robust to permutation-invariant noise injections and not rely on forms of redundancy such as positional embedding and input replications.
>
> These were also the reasons why we focused on making our comparisons with PointMamba instead of other works such as Point Cloud Mamba. In fact **PCM** presents many insightful experiments with different reordering strategies such as combining axis-wise reordering with other geospatial ordering such as Hiblert and z ordering, but **still has the two main weaknesses that our research tries to overcome**: the use of positional embedding and the replication of the input, which both can be considered as a form of redundancy fed to the model
>
> As also stated in our reply to reviewer 8URG, the choice of the threshold r is related to the intrinsic characteristics of the point clouds dataset used in similar works. **The value is a portion of the distance between the center and the border of the scene in which the object is inserted**. In our case, the value 0.8 gives a good balance between performance and computational cost.
>
> As also stated in our reply to reviewer 8URG, our time analysis is shown in Table 3. We make a comparison between **PointMamba, where the input sequence is tripled, and Nimba, which preserves the input sequence length**.

---

> > ### Comment · Reviewer_9zEE · 2024-11-26
> >
> > Although the difference of reordering strategies between PCM and this paper is discussed, the detailed comparison should present. And the other concerns mentioned above do not have solved.

---

### Official Review · Reviewer_8URG · 2024-11-01

**Soundness:** 3
**Presentation:** 2
**Contribution:** 2
**Rating:** 5
**Confidence:** 3

**Summary:**

This paper provides detailed introductions to existing Mamba-based point cloud analysis strategies. Then, this paper proposes a point cloud state space method named NIMBA to remove positional embedding and avoid data replication in related methods. Experiments demonstrate that the NIMBA outperforms PointMamba in robustness while facing spatial variations.

**Strengths:**

1. The target to remove positional embedding and avoid data replication is valuable.
2. The overall writing is fluent and clear.

**Weaknesses:**

1. Since point clouds are 3D data, would ordering them along a single axis be sub-optimal?
2. Point clouds are highly spatially scattered and disordered. A manually pre-defined $r$ may not be suitable for all scenes.
3. This paper can provide an inference time analysis to present an improvement in efficiency by avoiding data replication.

**Questions:**

1. Does Figure 2 illustrate that feeding wrong ordering centers to MAMBA Layers?

---

> ### Author Response · Authors · 2024-11-25
>
> We thank the reviewer for pointing out these issues with our submission:
>
> Regarding **using a y-axis ordering**, we want to highlight that this **is just the initial sequence from which Nimba’s reordering strategy will start** making swaps between patches.  Indeed, any reordering criteria could be used as an initial sequence and lead to the same final reordered sequence, but we empirically saw that starting with a y-axis reordering saved some computation for the next series of swaps. We suppose that this phenomenon is related to the natural orientation of the objects in the dataset.
>
> The choice of the threshold r to be 0.8, other than being a good balance between computation costs and performance, is related to the nature of point cloud datasets. Indeed, when dealing with highly spatial data modalities, a normalization step is always performed, resulting in the datasets ScanObject and ModelNet having objects in a [-1,1]3 cube. **The literature confirms that this is a standard procedure** in similar works and datasets. If the choice of r seems arbitrary, it can be interpreted as **just a portion of the distance between the center of the scene and the border of the scene**, which should be 80%.  We added this explanation to our paper.
>
> Regarding a time analysis inference comparison with replicated and not-replicated data, we showed this in Table 3. Indeed, the table shows a **comparison between PointMamba, which triplicates the sequence length, and Nimba, which keeps the input sequence length**. We want to highlight that this comparison takes into consideration both the preprocessing of the input (that encompasses the reordering strategies) and the actual training. We agree on the fact that the reordering shown in Figure 2 might be misleading, we will correct it accordingly.

---

### Official Review · Reviewer_Lgt4 · 2024-11-05

**Soundness:** 2
**Presentation:** 3
**Contribution:** 2
**Rating:** 5
**Confidence:** 5

**Summary:**

The paper introduces a improved approach to applying SSMs to 3D point cloud data. The study presents NIMBA, a Mamba-based model that uses a unique reordering strategy to convert 3D point clouds into 1D sequences while preserving spatial relationships, thus eliminating the need for positional embeddings and reducing data redundancy and computational overhead. Additionally, the NIMBA model demonstrates enhanced robustness to common data transformations.

**Strengths:**

1. The idea is simple and the paper is easy to follow.

2. Introducing Mamba to point clouds is non-trival and further improve the efficiency and robustness is important.

3. The analysis is soundness.

**Weaknesses:**

1. PointMamba, the article's main comparator, has been accepted by NeurIPS, and its methodology has been updated. The authors should revise the relevant descriptions in the article accordingly.

2. In the paper, the authors assert that NIMBA does not rely on positional embedding (PE). However, the ablation study in section 4.3.1 indicates that NIMBA performs better with PE, contradicting their claim (see line 97). Additionally, the results in Table 2 also seem to include PE, leading to confusion. Furthermore, since PE is easy to compute and doesn't significantly increase computational burden, the claim that NIMBA can contribute without it raises questions, especially given that omitting PE results in decreased performance

3. The reviewer is also unsure how NIMBA validates global modeling with a sequence length N in the causal modeling Mamba. While sequence order can help preserve geometric relationships, the point patches still struggle to interact with each other. More discussion can be added.

**Questions:**

See Weakness. Besides, there are few other suggestions:

1. Details in the writing require verification. For instance, the caption in Table 5 seems inaccurate; a full stop is missing at the end of lines 339 and 360.

2. The reviewers understand that the authors trained from scratch to better highlight the difference between each setting. However, stronger data augmentation and pre-training fine-tuning could still be added to demonstrate the upper limit of NIMBA's performance.

---

> ### Author Response · Authors · 2024-11-25
>
> We thank the reviewer for their comments on our submission:
>
> **We agree that comparisons should be ideally performed with the latest PointMaba version** (v4). The main reason why we used as a baseline an earlier version of PointMamba (v3) is that, as of today, **we have no code associated with** the reordering implemented in **PointMamba v4**, which leverages a Hilbert and transpose-Hilbert reordering strategy, doubling the original sequence length. However, we like to note that **we tried to implement** our version of this reordering strategy, **but we did not get the same performances shown in PointMamba v4**, so we suppose that also other changes were made that could unfortunately not be inferred from the sole text of the paper without the source code.
>
> We also want to highlight that we contacted the authors and we got confirmation that also **PointMamba v4 performance relies heavily on positional embedding**, showing the same conceptual weaknesses that inspired our work. What **our paper fundamentally shows is how to operate with point clouds beyond positional embedding and sequence replication** by utilizing a structure-aware input sequence reordering that allows robust global modeling. By operating beyond position embedding (PE), we mean that Nimba can preserve almost the same performances without PE (a gap of ~1%) and gains robustness towards permutation-invariant augmentation. Other models have a more significant drop in performance when removing PE (around ~4-6%) and are significantly less robust. Yet since PE tries to encode the relative position between points, adding it to the embedding of the patches gives a slight improvement in performance. However, this spatial information is already encoded in the patches themselves, since intuitively they represent actual coordinates. We however do not wish the model to rely on positional embeddings alone to navigate the point cloud, but to use the sequential structure imposed by Mamba processing: this inspired our investigation. **We agree that our claim in line 97 could be misleading, we propose to change** it from “without affecting” **to “without significantly affecting”**.
>
> We confirm that the caption in Table 5 is inaccurate, we will upload a correct version.
>
> **We decided to focus our experiments on models trained from scratch to better highlight what components are responsible for performances** when operating with point clouds (or more generally, unordered data modalities). Indeed, when using pre-trained architecture, these comparisons are harder to make and make a negligible improvement in performance (~1% in PointMamba v3).

---

### Meta-Review · Area_Chair_1aVz · 2024-12-19

**Metareview:**

In this paper, all reviewers vote for rejection. After checking the paper, the AC agreed to the rejection, considering the limited novelty and lacked details.

### Pros
1. **Efficient Design**: The proposed reordering strategy reduces computational costs and eliminates reliance on positional embeddings.
2. **Comprehensive Evaluation**: Demonstrates improvements in classification and segmentation tasks across multiple datasets.
3. **Robustness**: Shows resilience to noise and spatial transformations during testing.

### Cons
1. **Limited Novelty**: The main contribution is a reordering strategy, which lacks significant innovation beyond existing methods.
2. **Scalability Issues**: The model shows limited improvements when scaled or integrated with newer architectures like Mamba2.
3. **Narrow Focus**: Validation is restricted to a few standard datasets, limiting insights into generalizability.
4. **Integration Challenges**: Performance declines when the method is adapted into hybrid frameworks, suggesting integration issues.

The reason for rejection is as follows:

1. **Insufficient Innovation**: The contribution focuses on a reordering strategy, which, while effective, does not introduce significant methodological advancements.
2. **Scalability Concerns**: The limited scalability and integration challenges reduce the potential impact of the approach.
3. **Narrow Validation**: The lack of diverse datasets and tasks weakens the claim of generalizability.

**Additional Comments On Reviewer Discussion:**

Reviewers raised concerns about limited technical novelty and lack of experiments in the discussion. The author addresses those issues with more explanation but does not convince reviewers.

From the AC view, more experiments, especially the scaling of the model and data, should be conducted to prove the method's effectiveness.

The following is some details:

#### Points Raised by Reviewers
1. **Limited Novelty** (UZaR, TkwJ): The primary contribution, a reordering strategy, is incremental and lacks significant methodological innovation.
2. **Scalability Issues** (3fWp, Uzyh): Concerns were raised regarding the limited scalability of the approach, especially when applied to larger models or hybrid architectures.
3. **Narrow Evaluation Scope** (TkwJ, Uzyh): The evaluation was deemed insufficiently diverse, focusing primarily on standard datasets like ModelNet40 and ScanObjectNN.
4. **Integration Challenges** (3fWp, TkwJ): Performance declines when the approach is integrated with newer architectures, such as Mamba2.
5. **Efficiency vs. Accuracy Trade-off** (UZaR): Whether the efficiency gains justified the modest accuracy improvements compared to baseline models.

#### Author Responses
1. **Novelty**: The authors emphasized the importance of their reordering strategy for improving robustness and efficiency but acknowledged that the approach builds on existing SSM frameworks.
2. **Scalability**: The authors highlighted their model’s efficiency in smaller settings but did not provide substantial evidence to address scalability concerns.
3. **Evaluation Scope**: While the authors mentioned plans to extend their evaluation to diverse datasets, no additional results were provided during the rebuttal.
4. **Integration Challenges**: The authors attributed the performance declines to inherent limitations in newer architectures like Mamba2 and suggested future work to address this issue.
5. **Efficiency Gains**: The authors defended the trade-off by citing improved training efficiency and robustness but did not provide compelling evidence of substantial real-world impact.

#### Final Decision and Weighing of Points
The rebuttal provided clarifications but failed to address core concerns adequately:
1. **Limited Novelty**: The incremental nature of the contribution remains a critical weakness.
2. **Scalability and Generalizability**: The lack of additional evidence or experiments to demonstrate scalability and generalizability limited the impact of the approach.
3. **Integration Issues**: The unresolved challenges with hybrid architectures weaken the model’s applicability.
4. **Insufficient Improvements**: The efficiency gains, while notable, are not transformative enough to offset the modest accuracy improvements.

Given the unresolved concerns and limited contribution to the field, the decision to **reject** the paper was maintained.

---

### Decision · Program_Chairs · 2025-01-22

Reject